# Mixture Representation Learning with Coupled Autoencoders

## Abstract

Latent representations help unravel complex phenomena. While continuous latent variables can be efficiently inferred, fitting mixed discrete-continuous models remains challenging despite recent progress, especially when the discrete factor dimensionality is large. A pressing application for such mixture representations is the analysis of single-cell omic datasets to understand neuronal diversity and its molecular underpinnings. Here, we propose an unsupervised variational framework using multiple interacting networks called *cpl-mixVAE* that significantly outperforms state-of-the-art in high-dimensional discrete settings. *cpl-mixVAE* introduces a consensus constraint on discrete factors of variability across the networks, which regularizes the mixture representations at the time of training. We justify the use of this framework with theoretical results and validate it with experiments on benchmark datasets. We demonstrate that our approach discovers interpretable discrete and continuous variables describing neuronal identity in two single-cell RNA sequencing datasets, each profiling over a hundred cortical neuron types.

## 1 Introduction

Fitting mixed discrete-continuous models arises in many contexts. While continuous latent variables can be efficiently inferred with variational and adversarial formulations, inference of continuous and discrete factors in generalized mixture models remains challenging despite recent progress (Jang et al., 2016; Chen et al., 2016; Dupont, 2018; Jeong & Song, 2019). A pressing domain of application for such models is quantifying factors of biological variability in single-cell omic studies. The high-throughput and high-dimensional datasets produced by these studies document a previously unappreciated diversity of gene expression. In neuroscience, this poses the identification of cell types and cell states as a key research area to understand how neuronal circuits function (Bargmann et al., 2014), where the notions of cell types and states can be considered as biological interpretations of discrete and continuous variability. While marker gene based studies suggest the existence of more than 100 neuronal cell types in just a single brain region, there is no agreement on such categorization and interpretation of the remaining continuous variability (Seung & Sümbül, 2014; Zeng & Sanes, 2017; Tasic et al., 2018). Moreover, existing unsupervised, joint continuous-discrete learning methods are tailored for problems with relatively few and equally-abundant discrete components, and accurate inference remains out of reach for these applications.

Deep generative models have previously been applied to single-cell datasets, where the focus is on the cluster identity and it is typically inferred by post-hoc analysis of a continuous factor (Lopez et al., 2018). Deep Gaussian mixture models (Dilokthanakul et al., 2016; Johnson et al., 2016; Jiang et al., 2017) also focus on the identification of categories and do not take interpretability of the remaining continuous variability into account. To address the need for joint inference of interpretable discrete and continuous factors, various adversarial and variational methods have been proposed. While existing adversarial generative models, e.g. InfoGAN (Chen et al., 2016), are susceptible to stability issues (Higgins et al., 2017; Kim & Mnih, 2018), variational autoencoders (VAEs) (Kingma & Welling, 2013) emerge as efficient and more stable alternatives (Tschannen et al., 2018; Zhang et al., 2018; Dupont, 2018; Jeong & Song, 2019). VAE-based approaches approximate the mixture model by assuming a family of distributions $q_\phi$ and select the member closest to the true model $p$. Popular choices in VAE implementations include (1) using KL divergence to compute discrepancy between $q_\phi$ and $p$, and (2) using a multivariate Gaussian mixture distribution with uniformly distributed discrete and isotropic Gaussian distributed continuous priors. However, such choices may lead to

underestimating the posterior variance (Minka et al., 2005; Blei et al., 2017). Solutions to resolve this issue are mainly applicable in low-dimensional spaces or for continuous factors alone (Deasy et al., 2020; Kingma et al., 2016; Ranganath et al., 2016; Quiroz et al., 2018).

Inspired by *collective decision making*, we introduce a variational framework using multiple *autoencoding arms* to jointly infer interpretable finite discrete (categorical) and continuous factors in the presence of high-dimensional discrete space. Coupled-autoencoders have been previously studied in the context of multi-modal recordings, where each arm learns only a continuous latent representation for one of the data modalities (Feng et al., 2014; Gala et al., 2019; Lee & Pavlovic, 2020). Here, we develop a novel pairwise-coupled autoencoder framework for a single data modality. The proposed framework imposes a consensus constraint on the categorical posterior at the time of training and allows dependencies between continuous and categorical factors. We define the consensus constraint based on the Aitchison geometry in the probability simplex, which avoids the mode collapse problem. We show that the coupled multi-arm architecture enhances accuracy, robustness, and interpretability of the inferred factors without requiring any priors on the relative abundances of categories. Finally, on datasets profiling different cortical regions in the mammalian brain, we show that our method can be used to discover neuronal types as discrete categories and type-specific genes regulating the continuous within-type variability, such as metabolic state or disease state.

**Related work.** There is an extensive body of research on clustering in mixture models (Dilokthanakul et al., 2016; Jiang et al., 2017; Tian et al., 2017; Guo et al., 2016; Locatello et al., 2018b). The idea of improving the clustering performance through seeking a consensus and *co-training* and *ensembling* across multiple observations has been explored in both unsupervised (Monti et al., 2003; Kumar & Daumé, 2011) and semi-supervised contexts (Blum & Mitchell, 1998). However, these methods do not consider the underlying continuous variabilities across observations. Moreover, unlike ensemble methods, which pool the results of different trained workers, autoencoding arms seek a consensus at the time of learning in our framework.

The proposed framework does not need any supervision since the individual arms provide a form of prior or weak supervision for each other. In this regard, our paper is related to a body of work that attempts to improve representation learning by using semi-supervised or group-based settings (Bouchacourt et al., 2017; Hosoya, 2019; Nemeth, 2020). Bouchacourt et al. (2017) demonstrated a multi-level variational autoencoder (MLVAE) as a semi-supervised VAE by revealing that observations within groups share the same type. Hosoya (2019) and Nemeth (2020) attempted to improve MLVAE by imposing a weaker condition to the grouped data. In recent studies (Shu et al., 2019; Locatello et al., 2020), a weakly supervised variational setting has been proposed for disentangled representation learning by providing pairs of observations that share at least one underlying factor. These studies rely on learning latent variables in continuous spaces, and have been applied only to image datasets with low-dimensional latent representations.

Recent advances in structured variational methods, such as imposing a prior (Ranganath et al., 2016) or spatio-temporal dependencies (Quiroz et al., 2018) on the latent distribution parameters, allow for scaling to larger dimensions. However, these solutions are not directly applicable to the discrete space, which will be addressed in our A-arm VAE framework.

## 2 SINGLE MIXTURE VAE FRAMEWORK

For an observation $\mathbf{x} \in \mathbb{R}^D$, a VAE learns a generative model $p_{\boldsymbol{\theta}}(\mathbf{x}|\mathbf{z})$ and a variational distribution $q_{\boldsymbol{\phi}}(\mathbf{z}|\mathbf{x})$, where $\mathbf{z} \in \mathbb{R}^M$ is a latent variable with a parameterized distribution $p(\mathbf{z})$ and $M \ll D$ (Kingma & Welling, 2013). *Disentangling* different sources of variability into different dimensions of $\mathbf{z}$ enables an interpretable selection of latent factors (Higgins et al., 2017; Locatello et al., 2018a). However, the interplay between continuous and discrete variabilities present in many real-world datasets is often overlooked by existing methods. This problem can be addressed within the VAE framework in an unsupervised fashion by introducing a categorical latent variable $\mathbf{c}$ denoting the class label, alongside the continuous latent variable $\mathbf{s}$. We refer to the continuous variable $\mathbf{s}$ as the *state* or *style* variable interchangeably. Assuming $\mathbf{s}$ and $\mathbf{c}$ are independent random variables, the evidence lower bound (ELBO) (Blei et al., 2017) for a single mixture VAE with the distributions parameterized by $\boldsymbol{\theta}$ and $\boldsymbol{\phi}$ is given by,

$$\mathcal{L}(\boldsymbol{\phi}, \boldsymbol{\theta}) = \mathbb{E}_{q_{\boldsymbol{\phi}}(\mathbf{s},\mathbf{c}|\mathbf{x})}\left[\log p_{\boldsymbol{\theta}}(\mathbf{x}|\mathbf{s},\mathbf{c})\right] - D_{KL}\left(q_{\boldsymbol{\phi}}(\mathbf{s}|\mathbf{x})\|p(\mathbf{s})\right) - D_{KL}\left(q_{\boldsymbol{\phi}}(\mathbf{c}|\mathbf{x})\|p(\mathbf{c})\right). \quad (1)$$

Maximizing ELBO in Eq. 1 imposes characteristics on $q(\mathbf{s}|\mathbf{x})$ and $q(\mathbf{c}|\mathbf{x})$ that can result in underestimation of posterior probabilities such as the mode collapse problem, where the network ignores a

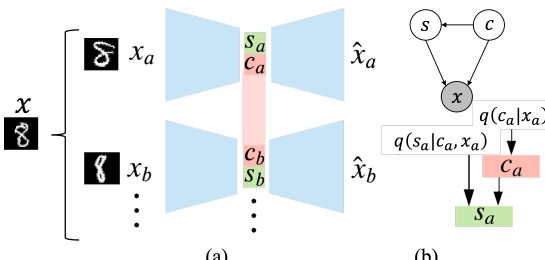

Figure 1: (a) Multi-arm autoencoder framework proposed as the cpl-mixVAE model. Individual arms receive non-identical noisy copies of given samples $\mathbf{x}$, i.e. $\{\mathbf{x}_a, \mathbf{x}_b, \dots\}$, where they all belong to the same category, to learn mixture representations, i.e. $\{q(\mathbf{c}_a, \mathbf{s}_a), q(\mathbf{c}_b, \mathbf{s}_b), \dots\}$. VAE arms cooperate to learn the categorical assignment, $p(\mathbf{c})$. (b) Graphical model of each autoencoder to learn type dependence of the state variable.

subset of latent variables (Minka et al., 2005; Blei et al., 2017). Recently, VAE-based solutions were proposed by imposing a uniform structure on $p(\mathbf{c})$: akin to $\beta$-VAE (Higgins et al., 2017; Burgess et al., 2018), JointVAE (Dupont, 2018) modifies the ELBO by assigning a pair of controlled information capacities for each variational factor, i.e. $\mathcal{C}_s \in \mathbb{R}^{|\mathbf{s}|}$ and $\mathcal{C}_c \in \mathbb{R}^{|\mathbf{c}|}$. The main drawback of JointVAE is that its performance is tied to heuristic tuning of $|\mathbf{s}| \times |\mathbf{c}|$ capacities over training iterations so that it is vulnerable to mode collapse in high-dimensional settings. Another recent VAE-based mixture model solution, CascadeVAE (Jeong & Song, 2019), maximizes the ELBO through a semi-gradient-based algorithm by iterating over two separate optimizations for the continuous and categorical variables. While the separation of the optimization steps avoids the mode collapse problem, this separation is valid only when the categorical variable is uniformly distributed. Therefore, its performance strongly depends on the clusters having similar abundances in the dataset. Thus, earlier solutions fall short of learning interpretable mixture representations with high-dimensional discrete variables in real-world applications.

In addition to the issues discussed above, the performance and interpretability of those approaches are further limited by the common assumption that the continuous variable representing the style of the data is independent of the categorical variable. In practice, style often depends on the class label. For instance, even for the well-studied MNIST dataset, the histograms of common digit styles, e.g. "width", markedly vary for different digits (Supplementary Section I). Moreover, further analysis of the identified continuous factor in the earlier approaches reveals that the independence assumption among $q(\mathbf{s}|\mathbf{x})$ and $q(\mathbf{c}|\mathbf{x})$ can be significantly violated (see Supplementary Sections H and I).

## 3    COUPLED MIXTURE VAE FRAMEWORK

The key intuition behind multi-arm networks is cooperation to improve posterior estimation. While the context is different, the popular phrase "wisdom of the crowd" (Surowiecki, 2005) can nevertheless be revealing: when a crowd (multiple arms) needs to make a decision, multiple estimates can increase the expected probability of a correct choice.

### 3.1    A-ARM VAE FRAMEWORK

We define the $A$-arm VAE as an $A$-tuple of independent and architecturally identical autoencoding arms, where the $a$-th arm parameterizes a mixture model distribution (Fig. 1a). In this framework, individual arms receive a collection of non-identical copies, $\{\mathbf{x}_a, \mathbf{x}_b, \dots\}$ of the given sample, $\mathbf{x}$, belonging to the same category. While each arm has its own mixture representation with potentially non-identical parameters, all arms cooperate to learn $q(\mathbf{c}_a|\mathbf{x}_a)$, where $\mathbf{c}_a = \mathbf{c}_b = \cdots$, via a cost function at the time of training. Accordingly, a crowd of VAEs with $A$ arms can be formulated as a collection of constrained variational objectives as follows.

$$\max \quad \mathcal{L}_{\mathbf{s}_1|\mathbf{c}_1}(\phi_1, \theta_1) + \cdots + \mathcal{L}_{\mathbf{s}_A|\mathbf{c}_A}(\phi_A, \theta_A) \tag{2}$$
$$\text{s.t. } \mathbf{c}_1 = \cdots = \mathbf{c}_A$$

where $\mathcal{L}_{\mathbf{s}_a|\mathbf{c}_a}(\phi_a, \theta_a)$ is the variational loss for arm $a$,

$$\mathcal{L}_{\mathbf{s}_a|\mathbf{c}_a}(\phi_a, \theta_a) = \mathbb{E}_{q(\mathbf{s}_a, \mathbf{c}_a|\mathbf{x}_a)}\left[\log p(\mathbf{x}_a|\mathbf{s}_a, \mathbf{c}_a)\right] - \mathbb{E}_{q(\mathbf{c}_a|\mathbf{x}_a)}\left[D_{KL}\left(q(\mathbf{s}_a|\mathbf{c}_a, \mathbf{x}_a)\|p(\mathbf{s}_a|\mathbf{c}_a))\right)\right]$$
$$- \mathbb{E}_{q(\mathbf{s}_a|\mathbf{c}_a, \mathbf{x}_a)}\left[D_{KL}\left(q(\mathbf{c}_a|\mathbf{x}_a)\|p(\mathbf{c}_a))\right)\right]. \tag{3}$$

In Eq. 3, the variational loss for each arm is defined according to the graphical model in Fig. 1b, which is built upon the traditional ELBO in Eq. 1 by conditioning the continuous state on the categorical variable (derivation in Supplementary Section B). Therefore, learning an interpretable decomposition of the data relies on accurate assignment (inference) of the categorical latent factor. Propositions 1 and 2 below show that the shared categorical assignment inferred from $q(\mathbf{c}|\mathbf{x}_1, \cdots, \mathbf{x}_A)$, under the

$\mathbf{c} = \mathbf{c}_1 = \cdots = \mathbf{c}_A$ constraint of the multi-arm framework improves the accuracy of the categorical assignment on expectation.

**Proposition 1.** *Consider the problem of mixture representation learning in a multi-arm VAE framework. For independent samples from category* $\mathbf{m}$*, i.e.* $\mathbf{x}_i \sim p(\mathbf{x}|\mathbf{m})$*,*

$$\mathbb{E}_{q(\mathbf{x}|\mathbf{m})}\left[\log q(\mathbf{c} = \mathbf{m}|\{\mathbf{x}_i\}_{1:A})\right] \quad > \quad \mathbb{E}_{q(\mathbf{x}|\mathbf{m})}\left[\log q(\mathbf{c} = \mathbf{m}|\{\mathbf{x}_i\}_{1:B})\right]$$
$$s.t.\ \mathbf{c} = \mathbf{c}_1 = \cdots = \mathbf{c}_A \qquad\qquad s.t.\ \mathbf{c} = \mathbf{c}_1 = \cdots = \mathbf{c}_B \tag{4}$$

*if* $q(\mathbf{m}|\mathbf{x}_i) < 1$ *and* $A > B \geq 1$ *denote the number of arms.* (Proof in Supplementary Section A)

Thus, having more arms increases the expected log posterior for the true categorical latent variable unless it is already at its maximum.

**Proposition 2.** *In the* $A$*-arm VAE framework, there exists an* $A$ *that guarantees a true categorical assignment on expectation. That is,*

$$\mathbf{m} = \arg\max_{\mathbf{c}} \mathbb{E}_{q(\mathbf{x}|\mathbf{m})}\left[\log q(\mathbf{c}|\{\mathbf{x}_i\}_{1:A})\right], \quad s.t.\ \mathbf{c} = \mathbf{c}_1 = \cdots = \mathbf{c}_A\ . \tag{5}$$

(Proof in Supplementary Section A)

Accordingly, the consensus constraint is sufficient to enhance inference for mixture representations in the $A$-arm VAE framework. Our theoretical results show that the required number of arms satisfying Eq. 5 is a function of the categorical distribution and the likelihood (Eq. 15, Supplementary Section A). In the particular case of uniformly distributed categories, one pair of coupled arms is enough to satisfy Eq. 5 (see Corollary 1, Supplementary Section A).

We emphasize that the proposed framework does not require any weak supervision as in (Bouchacourt et al., 2017). Instead, it relies on representations that are invariant under non-identical copies of observations. Moreover, unlike (Bouchacourt et al., 2017; Shu et al., 2019; Locatello et al., 2020), the multi-arm framework is not restricted to the continuous space.

**Arms observe non-identical copies of samples.** In the A-arm VAE framework, arms receive non-identical observations that share the discrete variational factor. To achieve this in a fully unsupervised setting, we use *type-preserving* data augmentation that generates independent and identically distributed copies of data while preserving its categorical identity. For image datasets, conventional transformations such as rotation, scaling, or translation can serve as type-preserving augmentations. However, for non-image datasets, e.g. single-cell data, we seek a generative model that learns transformations representing within-class variability in an unsupervised manner. To this end, inspired by DAGAN (Antoniou et al., 2017) and VAE-GAN (Larsen et al., 2016), we develop a generative model to provide collections of observations for our multi-arm framework (Supplementary Section F). The proposed generative model learns to generate augmented samples in the vicinity of given samples in the latent space, without knowing their types (Eq. 70). In Supplementary Section A, Remark 2, we further discuss an under-exploration scenario in data augmentation, in which the augmented samples are not independently distributed and are concentrated around the given sample.

### 3.2 CPL-MIXVAE: PAIRWISE COUPLING IN A-ARM VAE

In the $A$-arm VAE framework, the mixture representation is obtained through the optimization in Eq. 2. Not only is it challenging to solve the maximization in Eq. 2 due to the equality constraint, but the objective remains a function of $p(\mathbf{c})$ which is unknown, and typically non-uniform. To overcome this, we use an equivalent formulation for Eq. 2 by applying the pairwise coupling paradigm as follows (details of derivation in Supplementary Section C):

$$\max \quad \sum_{a=1}^{A} (A-1)\left(\mathbb{E}_{q(\mathbf{s}_a,\mathbf{c}_a|\mathbf{x}_a)}\left[\log p(\mathbf{x}_a|\mathbf{s}_a,\mathbf{c}_a)\right] - \mathbb{E}_{q(\mathbf{c}_a|\mathbf{x}_a)}\left[D_{KL}\left(q(\mathbf{s}_a|\mathbf{c}_a,\mathbf{x}_a)\|p(\mathbf{s}_a|\mathbf{c}_a))\right)\right]\right) -$$
$$\sum_{a<b} \mathbb{E}_{q(\mathbf{s}_a|\mathbf{c}_a,\mathbf{x}_a)}\mathbb{E}_{q(\mathbf{s}_b|\mathbf{c}_b,\mathbf{x}_b)}\left[D_{KL}\left(q(\mathbf{c}_a|\mathbf{x}_a)q(\mathbf{c}_b|\mathbf{x}_b)\|p(\mathbf{c}_a,\mathbf{c}_b))\right)\right]$$
$$s.t.\ \mathbf{c}_a = \mathbf{c}_b\ \forall a,b \in [1,A],\ a < b \tag{6}$$

We relax the optimization in Eq. 6 into an unconstrained problem by marginalizing the joint distribution over a mismatch measure between categorical variables (see Supplementary Section D):

$$\max \sum_{a=1}^{A} (A-1)\left(\mathbb{E}_{q(\mathbf{s}_a,\mathbf{c}_a|\mathbf{x}_a)}\left[\log p(\mathbf{x}_a|\mathbf{s}_a,\mathbf{c}_a)\right] - \mathbb{E}_{q(\mathbf{c}_a|\mathbf{x}_a)}\left[D_{KL}\left(q(\mathbf{s}_a|\mathbf{c}_a,\mathbf{x}_a)\|p(\mathbf{s}_a|\mathbf{c}_a))\right)\right]\right) +$$
$$\sum_{a<b} H(\mathbf{c}_a|\mathbf{x}_a) + H(\mathbf{c}_b|\mathbf{x}_b) - \lambda\mathbb{E}_{q(\mathbf{c}_a,\mathbf{c}_b|\mathbf{x}_a,\mathbf{x}_b)}\left[d^2(\mathbf{c}_a,\mathbf{c}_b)\right] \tag{7}$$

In Eq. 7, in addition to entropy-based confidence penalties known as mode collapse regularizers (Pereyra et al., 2017), the distance measure $d(\mathbf{c}_a, \mathbf{c}_b)$ encourages a consensus on the categorical assignment controlled by $\lambda \geq 0$, the coupling hyperparameter.

We refer to the model in Eq. 7 as *cpl-mixVAE* (Fig. 1a). In cpl-mixVAE, VAE arms try to achieve identical categorical assignments while independently learning their own style variables. In experiments, we set $\lambda = 1$ universally. While the bottleneck architecture already encourages interpretable continuous variables, this formulation can be easily extended to include an additional hyperparameter to promote disentanglement of continuous variables as in $\beta$-VAE (Higgins et al., 2017). Additional analyses to assess the sensitivity of the cpl-mixVAE's performance to its coupling factor can be found in the Supplementary Section G.

It may be instructive to cast Eq. 7 in an equivalent constrained optimization form.

**Remark 1.** *The A-arm VAE framework is a collection of constrained variational models as follows:*

$$\max \sum_{a=1}^{A} \mathbb{E}_{q(\mathbf{s}_a, \mathbf{c}_a | \mathbf{x}_a)} \left[ \log p(\mathbf{x}_a | \mathbf{s}_a, \mathbf{c}_a) \right] - \mathbb{E}_{q(\mathbf{c}_a | \mathbf{x}_a)} \left[ D_{KL} \left( q(\mathbf{s}_a | \mathbf{c}_a, \mathbf{x}_a) \| p(\mathbf{s}_a | \mathbf{c}_a) \right) \right] + H(\mathbf{c}_a | \mathbf{x}_a)$$
$$s.t.\ \mathbb{E}_{q(\mathbf{c}_a | \mathbf{x}_a)} \left[ d^2(\mathbf{c}_a, \mathbf{c}_b) \right] < \epsilon \tag{8}$$

*where $\epsilon$ denotes the strength of the consensus constraint. Here, $\mathbf{c}_b$ indicates the assigned category by any one of the arms, $b \in \{1, \ldots, A\}$, imposing structure on the discrete variable to approximate its prior distribution.*

**Distance between categorical variables.** $d(\mathbf{c}_a, \mathbf{c}_b)$ denotes the distance between a pair of $|\mathbf{c}|$-dimensional un-ordered categorical variables, which are associated with probability vectors with non-negative entries and sum-to-one constraint that form a $K$-dimensional simplex, where $K = |\mathbf{c}|$. In the real space, a typical choice to compute the distance between two vectors is using Euclidean geometry. However, this geometry is not suitable for probability vectors. Here, we utilize *Aitchison geometry* (Aitchison, 1982; Egozcue et al., 2003), which defines a vector space on the simplex. Accordingly, the distance in the simplex, i.e. $d_{S^K}(\mathbf{c}_a, \mathbf{c}_b)$ is defined as $d_{S^K}(\mathbf{c}_a, \mathbf{c}_b) = \|clr(\mathbf{c}_a) - clr(\mathbf{c}_b)\|_2, \forall \mathbf{c}_a, \mathbf{c}_b \in \mathcal{S}^K$, where $clr(\cdot)$ denotes the *isometric centered-log-ratio* transformation in the simplex. This categorical distance satisfies the conditions of a mathematical metric according to Aitchison geometry.

### 3.3 SEEKING CONSENSUS IN THE SIMPLEX

An instance of the mode collapse problem (Lucas et al., 2019) manifests itself in the minimization of $d_{S^K}(\mathbf{c}_a, \mathbf{c}_b)$ (Eq. 7): its trivial local optima encourages the network to abuse the discrete latent factor by ignoring many of the available categories. In the extreme case, the representations can collapse onto a single category; $\mathbf{c}_a = \mathbf{c}_b = \mathbf{c}_0$. In this scenario, the continuous variable is compelled to act as a primary latent factor, while the model fails to deliver an interpretable mixture representation despite achieving an overall low loss value. To avoid such undesirable local equilibria while training, we add perturbations to the categorical representation of each arm. If posterior probabilities in the simplex have small dispersion, the perturbed distance calculation overstates the discrepancies. Thus, instead of minimizing $d_{S^K}^2(\mathbf{c}_a, \mathbf{c}_b)$, we minimize a perturbed distance $d_\sigma^2(\mathbf{c}_a, \mathbf{c}_b) = \sum_k \left( \sigma_{a_k}^{-1} \log c_{a_k} - \sigma_{b_k}^{-1} \log c_{b_k} \right)^2$, which corresponds to the distance between additively perturbed $\mathbf{c}_a$ and $\mathbf{c}_b$ vectors in Aitchison geometry. Here, $\sigma_{a_k}^2$ and $\sigma_{b_k}^2$ indicate the mini-batch variances of the $k$-th category, for arms $a$ and $b$. We next show that the perturbed distance $d_\sigma(\cdot)$ is bounded by $d_{S^K}(\cdot)$ and non-negative values $\rho_u, \rho_l$:

**Proposition 3.** *Suppose $\mathbf{c}_a, \mathbf{c}_b \in \mathcal{S}^K$, where $\mathcal{S}^K$ is a simplex of $K > 0$ parts. If $d_{S^K}(\mathbf{c}_a, \mathbf{c}_b)$ denotes the distance in Aitchison geometry and $d_\sigma^2(\mathbf{c}_a, \mathbf{c}_b) = \sum_k \left( \sigma_{a_k}^{-1} \log c_{a_k} - \sigma_{b_k}^{-1} \log c_{b_k} \right)^2$ denotes a perturbed distance, then*

$$d_{S^K}^2(\mathbf{c}_a, \mathbf{c}_b) - \rho_l \leq d_\sigma^2(\mathbf{c}_a, \mathbf{c}_b) \leq d_{S^K}^2(\mathbf{c}_a, \mathbf{c}_b) + \rho_u$$

*where $\rho_u, \rho_l \geq 0$, $\rho_u = K \left( \tau_{\sigma_u}^2 + \tau_{\mathbf{c}}^2 \right) + 2\Delta_\sigma \tau_{\mathbf{c}}$, $\rho_l = \dfrac{\Delta_\sigma^2}{K} - K\tau_{\sigma_l}^2$, $\tau_{\mathbf{c}} = \max_k \{ \log c_{a_k} - \log c_{b_k} \}$, $\tau_{\sigma_u} = \max_k \{ g_k \}$, $\tau_{\sigma_l} = \max_k \{ -g_k \}$, $\Delta_\sigma = \sum_k g_k$, and $g_k = (\sigma_{a_k}^{-1} - 1) \log c_{a_k} - (\sigma_{b_k}^{-1} - 1) \log c_{b_k}$. (Proof in Supplementary Section E)*

Thus, when $\mathbf{c}_a$ and $\mathbf{c}_b$ are similar and their spread is not small, $d_\sigma(\mathbf{c}_a, \mathbf{c}_b)$ closely approximates $d_{S^\kappa}(\mathbf{c}_a, \mathbf{c}_b)$. Otherwise, it diverges from $d_{S^\kappa}(\cdot)$ to avoid mode collapse.

## 4 EXPERIMENTS

We used four datasets: dSprites, MNIST, and two single-cell RNA sequencing (scRNA-seq) datasets; Smart-seq ALM-VISp (Tasic et al., 2018) and 10X MOp (Yao et al., 2021). Although dSprites and MNIST datasets do not require high-dimensional settings for mixture representation, to facilitate comparisons of cpl-mixVAE with earlier methods, first we report the results for these benchmark datasets. We trained three unsupervised VAE-based methods for mixture modeling: JointVAE (Dupont, 2018), CascadeVAE (Jeong & Song, 2019), and ours (cpl-mixVAE). For MNIST, we additionally trained the popular InfoGAN (Chen et al., 2016) as the most comparable GAN-based model. To show the interpretability of the mixture representations, (i) for the discrete latent factor, we report the accuracy (ACC) of categorical assignments and the $D_{KL}(q(\mathbf{c})\|p(\mathbf{c}))$, (ii) for the continuous variable, we perform latent traversal analysis by fixing the discrete factor and changing the continuous variable according to $p(\mathbf{s}|\mathbf{c}, \mathbf{x})$. We calculated the accuracy by using minimum weight matching (Kuhn, 1955) to match the categorical variables obtained by cpl-mixVAE with the available cluster labels for each dataset. Additionally, we report the computational efficiency (number of iterations per second) to compare the training complexity of the multi-arm framework against earlier methods (Table 1). All reported numbers for cpl-mixVAE models are average accuracies calculated across arms. In VAE-based models, to sample from $q(\mathbf{c}_a|\mathbf{x}_a)$, we use the Gumbel-softmax distribution (Jang et al., 2016; Maddison et al., 2014). In cpl-mixVAE, each arm received an augmented copy of the original input generated by the deep generative augmenter (Supplementary Section F) during training. Details of the network architectures and training settings can be found in Supplementary Section L.

### 4.1 BENCHMARK DATASETS

**dSprites.** dSprites is procedurally generated from discrete (3 shapes) and continuous (6 style factors: scale, rotation, and position) latent factors. Based on the uniform distribution of classes, we used a 2-arm cpl-mixVAE with $|\mathbf{c}| = 3$ and $|\mathbf{s}| = 6$ to learn interpretable representations. Results in Table 1 show that our method outperforms the other methods in terms of categorical assignment accuracy. In addition to demonstrating the traversal results (Fig. 2, bottom row), we report disentanglement scores (DS in Table 1). Even though the continuous factors do not depend on the discrete factors in this synthetic dataset, we did not change the architecture and expected the network to infer this independence. For a fair comparison, we used the same disentanglement metric implemented for CascadeVAE (Jeong & Song, 2019).

**MNIST.** Similarly, due to the uniform distribution of digit labels in MNIST, we again used a 2-arm cpl-mixVAE model. Following the convention (Dupont, 2018; Jeong & Song, 2019; Bouchacourt

Table 1: Training results for all datasets. cpl-mixVAE uses 2 arms. $|\mathbf{c}|$ and $|\mathbf{s}|$ denote the cardinality of latent discrete and continuous spaces. Iter, ACC and DS denote number of iterations, the accuracy of the categorical assignment, and the disentanglement score, respectively. Computation denotes the training speed (iteration/second) on a GeForce RTX 2080 Ti GPU. The computation of cpl-mixVAE includes the entire execution time for training one pair of coupled networks, plus data augmentation.

| Dataset | Iter | $|\mathbf{c}|$ | $|\mathbf{s}|$ | Method | ACC (%) $\uparrow$ (mean $\pm$ s.d.) | Computation $\uparrow$ (iteration/sec) | |
|---|---|---|---|---|---|---|---|
| dSprites | 300K | 3 | 6 | JointVAE | $44.79 \pm 03.9$ | 52.6 | $74.5 \pm 5.2$ |
| | | | | CascadeVAE | $78.84 \pm 15.7$ | 15.4 | $90.5 \pm 5.3$ |
| | | | | cpl-mixVAE | $\mathbf{96.30 \pm 09.2}$ | 20.6 | $89.9 \pm 4.1$ |
| MNIST | 120K | 10 | 2 | InfoGAN | $77.87 \pm 21.7$ | 12.2 | $2.15 \pm 4.2$ |
| | | 10 | 10 | JointVAE | $68.99 \pm 11.8$ | 74.1 | $3.12 \pm 1.0$ |
| | | | | JointVAE$^\dagger$ | $68.21 \pm 09.6$ | 54.4 | $3.65 \pm 2.7$ |
| | | | | CascadeVAE | $74.83 \pm 06.9$ | 23.8 | $1.99 \pm 1.1$ |
| | | | | CascadeVAE$^\dagger$ | $72.98 \pm 13.4$ | 17.8 | $2.44 \pm 1.4$ |
| | | | | cpl-mixVAE | $\mathbf{84.56 \pm 06.5}$ | 17.5 | $\mathbf{1.48 \pm 1.4}$ |
| Smart-seq ALM-VISp | 40K | 115 | 2 | JointVAE | $12.53 \pm 02.9$ | 28.6 | $459 \pm 12$ |
| | | | | CascadeVAE | $02.69 \pm 00.1$ | 03.4 | $54.8 \pm 0.5$ |
| | | | | cpl-mixVAE | $\mathbf{38.78 \pm 01.3}$ | 10.1 | $\mathbf{43.9 \pm 3.5}$ |
| 10X MOp | 220K | 140 | 2 | JointVAE | $02.32 \pm 00.1$ | 14.8 | $542 \pm 17$ |
| | | | | CascadeVAE | $02.85 \pm 00.5$ | 00.6 | $150 \pm 3.5$ |
| | | | | cpl-mixVAE | $\mathbf{23.64 \pm 00.9}$ | 08.2 | $\mathbf{135 \pm 5.2}$ |

(DS for dSprites; $D_{KL}(q(\mathbf{c})\|p(\mathbf{c})) (\times 100)$ for the others)

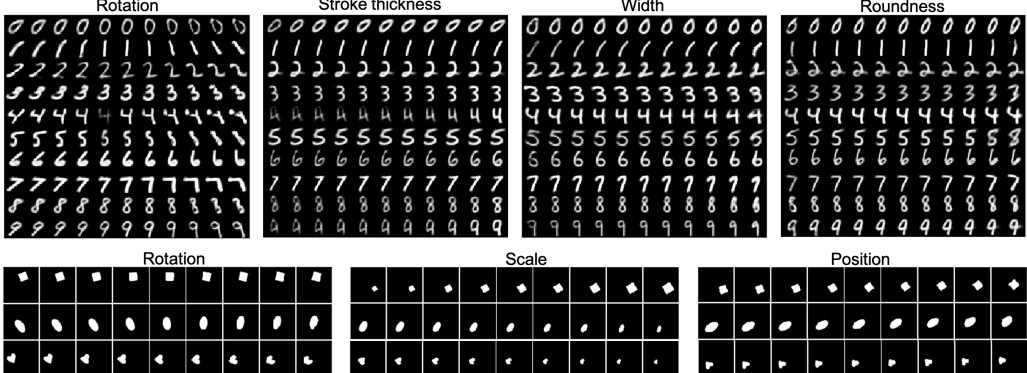

Figure 2: Interpretable continuous latent traversals of 1-st arm of the cpl-mixVAE framework with two autoencoders, for MNIST (top) and dSprites (bottom). The discrete variable **c** is constant for all reconstructions in the same row.

et al., 2017), each arm of cpl-mixVAE uses a 10-dimensional categorical variable representing digits (type), and a 10-dimensional continuous random variable representing the writing style (state). Table 1 displays the accuracy of the categorical assignment and the discrepancy between $q(\mathbf{c})$ and $p(\mathbf{c})$ for InfoGAN, two 1-arm VAE methods (JointVAE and CascadeVAE), and cpl-mixVAE with 2 arms. Additionally, to isolate the impact of data augmentation in training, we trained JointVAE[†] and CascadeVAE[†] where the models were trained with the same augmented copies of the original MNIST dataset as cpl-mixVAE. The results in Table 1 suggest that data augmentation by itself does not enhance the performance. Fig. 2 (top row) illustrates the continuous latent traversals for four dimensions of the state variable inferred by cpl-mixVAE, where each row corresponds to a different dimension of the categorical variable, and the state variable monotonically changes across columns. Both results in Table 1 and Fig. 2 show that cpl-mixVAE achieved an interpretable mixture representation with the highest categorical assignment accuracy.

**Summary.** cpl-mixVAE improves the discrete density approximation and infers better mixture representations. It outperforms earlier methods, without using extraneous optimization or heuristic channel capacities. Beyond performance and robustness, its computational cost is also comparable to that of the baselines.

## 4.2 SINGLE-CELL RNA SEQUENCING DATA

In this dataset the observations are individual cells and each observation consists of expressions of thousands of genes. Here, we used two scRNA-seq datasets: (i) Smart-seq ALM-VISp (Tasic et al., 2018) and (ii) 10X MOp (Yao et al., 2021). The Smart-seq dataset includes transcriptomic profiles of more than $10,000$ genes for $\sim 22,000$ cells from the mouse anterior lateral motor cortex (ALM) and the primary visual cortex (VIPs). The 10X MOp dataset profiles $\sim 123000$ cells in the mouse primary motor cortex (MOp) with the droplet-based 10X Genomics Chromium platform. 10X-based data often display more gene dropouts, especially for genes with lower expression levels. scRNA-seq datasets are significantly more complex than typical benchmark datasets due to (i) large number of cell types (discrete variable), and (ii) class imbalance; in the 10X MOp dataset, for instance, the most- and the least-abundant cell types include $17,000$ and $20$ samples, respectively. Moreover, whether the observed diversity corresponds to discrete variability or a continuum is an ongoing debate in neuroscience (Scala et al., 2020). While using genes that are differentially expressed in subsets of cells, known as *marker genes* (MGs) (Trapnell, 2015) is a common approach to define cell types, the identified genes rarely obey the idealized MG definition in practice. Here, we focus on *neuronal* cells and use a subset of $5,000$ highest variance genes. The original MG-based studies for each dataset suggested $115$ (Smart-seq data) (Tasic et al., 2018) and $140$ (10X-based data) (Yao et al., 2021) discrete neuronal types.

**Neuron type identification.** Based on the suggested taxonomies in (Tasic et al., 2018; Yao et al., 2021), for the Smart-seq ALM-VISp data, we used 115- and 2-dimensional discrete and continuous variables, and for the 10X MOp data, we used 140- and 2-dimensional discrete and continuous latent variables. We compared the suggested cell types in (Tasic et al., 2018; Yao et al., 2021) with the discrete representations that are inferred from VAE models. Table 1 and Fig. 3(a-b) demonstrate the

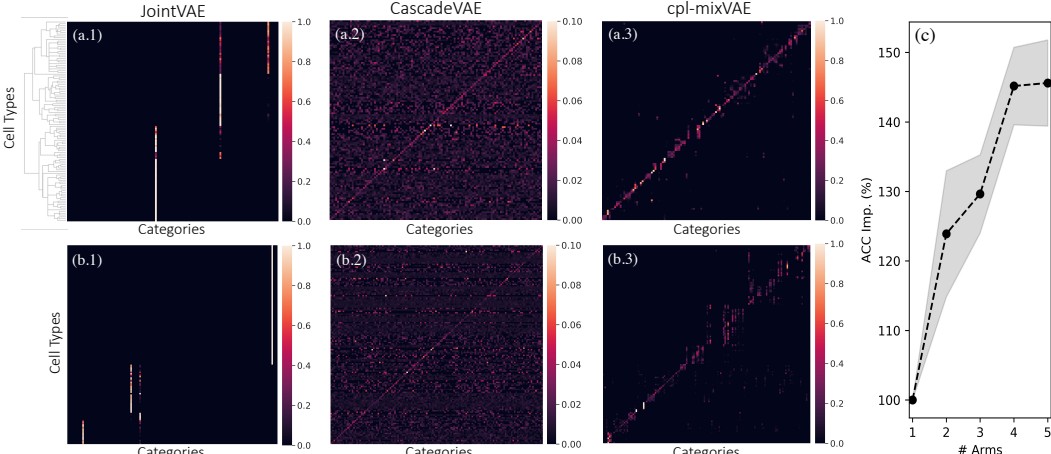

Figure 3: Categorical assignments for the scRNA-seq datasets. The top row shows confusion matrices of JointVAE (a.1), CascadeVAE (a.2), and cpl-mixVAE (a.3) trained by $|\mathbf{c}| = 115$, $|\mathbf{s}| = 2$, for the Smart-seq ALM-VISp dataset. The dendrogram on the y-axis shows MG-based hierarchical classification with 115 cell types, suggested by Tasic et al. (2018). The bottom row shows confusion matrices of JointVAE (b.1), CascadeVAE (b.2), and cpl-mixVAE (b.3) trained by $|\mathbf{c}| = 140$, $|\mathbf{s}| = 2$, for the 10X MOp dataset. Cell types on the y-axis are sorted based on a hierarchical classification suggested by Yao et al. (2021). (c) Improvement of the categorical representation (ACC) of cpl-mixVAE by adding more arms to the multi-arm framework. $A$-arm's performance for $A \geq 2$ is compared with the baseline 1-arm, JointVAE, across 3 randomly initialized runs.

performance of a 2-arm cpl-mixVAE model against JointVAE and CascadeVAE. In Fig. 3a.1 and 3b.1, we observe that for both datasets, JointVAE succeeds in identifying (sub)classes of neurons, e.g. excitatory class, or Sst subclass, but not neuronal types at leaf nodes. On the other hand, CascadeVAE learns an almost uniform distribution over all types despite a sizeable difference between the relative abundances of neuronal types (Fig. 3a.2 and 3b.2). Our results in Fig. 3(a-b) clearly show that cpl-mixVAE outperforms JointVAE and CascadeVAE in identifying meaningful known cell types. The confusion matrices in Fig. 3a.3 and 3b.3 demonstrate that even the inaccurate categorical assignments of cpl-mixVAE are still close to the matrix diagonals, suggesting a small cophenetic distance. That is, those cells are still assigned to nearby cell types in the dendrogram.

**Using A > 2.** Unlike the discussed benchmark datasets, the neuronal types are not uniformly distributed. Accordingly, we also investigated the accuracy improvement for categorical assignment when more than two arms are used. Fig. 3c illustrates the accuracy improvement with respect to a single autoencoder model, i.e. JointVAE, in agreement with our theoretical findings.

**Identifying genes regulating cell activity.** To examine the role of the continuous latent variable, we applied a similar traversal analysis to that used for the benchmark datasets. For a given cell sample and its discrete type, we changed each dimension of the continuous variable using the conditional distribution, and inspected gene expression changes caused by continuous variable alterations. Fig. 4 shows the results of the continuous traversal study for JointVAE and cpl-mixVAE, for two excitatory neurons belonging to the "L5 NP" (cell type (I)) and "L6 CT" (cell type (II)) sub-classes in ALM and MOp regions. Note that here, JointVAE is equivalent to a 1-arm VAE, with the exception of the type dependence of the state variable. Since CascadeVAE did not learn meaningful clustering of cells, even at the subclass level, we did not consider it for the continuous factor analysis. In each sub-figure, the latent traversal is color-mapped to normalized reconstructed expression values, where the $y$-axis corresponds to one dimension of the continuous variable, and the $x$-axis corresponds to three gene subsets, namely (i) MGs for the two excitatory types, (ii) immediate early genes (IEGs), and (iii) housekeeping gene (HKG) subgroups (Hrvatin et al., 2018; Tarasenko et al., 2017). For cpl-mixVAE (Fig. 4b), the normalized expression of the reported MGs as indicators for excitatory cell types (discrete factors) is unaffected by changes of identified continuous variables. In contrast, for JointVAE (Fig. 4a), we observed that the normalized expression of some MGs (5 out of 10) are changed due to the continuous factor traversal. Additionally, we found that the expression changes inferred by cpl-mixVAE for IEGs and HKGs are essentially *monotonically* linked to the continuous variable, confirming that the expression of IEGs and HKGs depends strongly on the cell activity variations under different metabolic and environmental conditions. Conversely, JointVAE

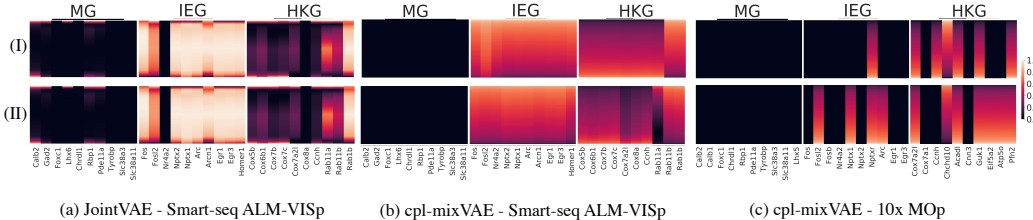

Figure 4: Continuous latent traversal analysis for two excitatory cell types: (I) "L5 NP" and (II) "L6 CT", in different brain regions. For each type, the traversal is color-mapped to a normalized reconstructed gene expression value (colorbar) as a function of the state variable for 3 gene subsets: marker genes (MG), immediate early genes (IEG), and housekeeping genes (HKG).

fails to reveal such activity-regulated monotonicity for IEGs and HKGs. Furthermore, our results for cpl-mixVAE reveal that the expression of activity-regulated genes depends on the cell type, i.e. IEGs and HKGs respond differently to activation depending on their cell types (compare rows I and II in Fig. 4b). However, in Fig. 4a, since the baseline JointVAE does not take into account the dependency of discrete and continuous factors, it fails to reveal the dependence of activity-regulated expression to the cell type, and therefore produces identical expressions for both types (I) and (II). These findings are consistent over multiple randomly initialized runs (Supplementary Section K.1). See Supplementary Section K.2 for more results on other cell types and gene subsets.

**Summary.** The cpl-mixVAE model successfully identified the majority of known excitatory and inhibitory neurons in multiple cortical regions. Our findings suggest that cpl-mixVAE, by acknowledging the dependencies of continuous and categorical factors, captures relevant and interpretable continuous variability that can provide insight when deciphering the molecular mechanisms shaping the landscape of biological states, e.g. due to metabolism or disease.

### 4.3    ABLATION STUDIES

To elucidate the success of the A-arm VAE framework in mixture modeling, we investigate the categorical assignment performance under different training settings. Since CascadeVAE does not learn the categorical factors by variational inference, here we mainly study JointVAE (as a 1-arm VAE) and cpl-mixVAE (as a 2-arm VAE). In Section 4.1, we show that data augmentation by itself does not enhance the categorical assignment (JointVAE[†]). To understand whether architectural differences put JointVAE at a disadvantage, we trained JointVAE[‡] (Table S1), which uses the same architecture as the one used in cpl-mixVAE. JointVAE[‡] uses the same learning procedure as JointVAE, but its convolutional layers are replaced by fully-connected layers (see Supplementary Section J and L for details). The result for JointVAE[‡] suggests that the superiority of cpl-mixVAE is not due to the network architecture either. We also examined the performance changes of the proposed 2-arm cpl-mixVAE under three different settings: (i) cpl-mixVAE$^*$, where coupled networks are not independent and network parameters are shared; (ii) cpl-mixVAE$^a$, where only affine transformations are used for data augmentation; and (iii) cpl-mixVAE$(\mathbf{s} \not\mid \mathbf{c})$, where the state variable is independent of the discrete variable (Table S1). Our results show that the proposed cpl-mixVAE obtained the best categorical assignments across all training settings. We also examined the accuracy of categorical assignments for the cpl-mixVAE model, under different dimensions of discrete latent variable, for both MNIST and scRNA-seq datasets (see Supplementary Section J). We experimentally observe that while JointVAE suffers from sensitivity to empirical choices of $|\mathbf{c}|$, cpl-mixVAE is more robust in encoding the discrete variability, without suffering from mode collapse (Fig. S9 and Fig. S10).

## 5    CONCLUSION

We have proposed cpl-mixVAE as a multi-arm framework to apply the power of collective decision making in unsupervised joint representation learning of discrete and continuous factors, scalable to the high-dimensional discrete space. This framework utilizes multiple pairwise-coupled autoencoding arms with a shared categorical variable, while independently learning the continuous variables. Our experimental results for all datasets support the theoretical findings, and show that cpl-mixVAE outperforms comparable models. Importantly, for challenging scRNA-seq dataset, we showed that the proposed framework identifies biologically interpretable cell types and differentiate between type-dependent and activity-regulated genes.

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
