# OpenReview forum: "Mixture Representation Learning with Coupled Autoencoders"
_ICLR.cc/2022/Conference — ICLR 2022 Submitted_

### Official Review · Reviewer_2RLY · 2021-10-24

**Correctness:** 4
**Technical Novelty And Significance:** 2
**Empirical Novelty And Significance:** 2
**Recommendation:** 5
**Confidence:** 3

**Main Review:**

The paper compares the proposed method with a few similar methods, using MNIST but also using a dataset for single-cell RNA sequencing data (a domain with which I am not familiar). I am also not familiar with how much efforts are currently being put into this approach of factorizing Z=(S,C) by other members in the field.

My main question for the authors is to ask about the results in table 1 when it comes to MNIST: do the std measurements indicate a spread over the multiple experimental runs, and why is the accuracy so low on MNIST? MNIST is the kind of dataset where a linear classifier can get 90% accuracy, yet all these methods presented fare much worse.

Minor issue: I think it would be appropriate to mention Kingma & Welling in the background section when VAE are first mentioned in the paper instead of in the later Section 2 when they are described. The same paper can be referenced twice, but at least it should be referenced in the introduction in conjunction to the other papers on VAEs that the authors wish to mention.

**Summary Of The Paper:**

The authors consider the problem of factorization of the hidden space of a VAE into two separate components Z=(S,C) where S is continuous and C is discrete. They make a particular assumption about the factorization of the encoder function q(S, C|X)=q(S|X)q(C|X) and they also take a mixture of “A” experts that are made to agree with their discrete assignment C.

**Summary Of The Review:**

Hard to say how novel the contribue is. I want to see the author respond to my question about table 1.

---

> ### Author Response · Authors · 2021-11-13
> **Response to reviewer 2RLY**
>
> We thank the reviewer for their feedback.
>
> **Main questions:**
>
> - Yes, the std measurements indicate the spread over 10 randomly initialized runs.
>
> - Regarding MNIST, we believe the review confuses unsupervised learning (clustering) vs supervised learning (classification) studies. In classification studies, the correct label is provided for training data and, indeed, simple classifiers can achieve ~90% accuracy in this (easy) regime. However, in unsupervised learning, no label information is provided to guide the training process, which represents a significantly harder problem. In unsupervised mixture modeling, the baselines reported here represent the state-of-the-art for joint, interpretable inference of categorical and continuous factors of variability. In this setting, our framework represents the best performance with significant improvements, particularly for the high dimensional discrete settings.
>
> **Minor issue:**
>
> - We added an extra citation to Kingma & Welling in Introduction (paragraph 2, line 8).
>
>
>
> We will be happy to address any further concerns the reviewer may have.

---

> > ### Author Response · Authors · 2021-11-27
> > **Follow up**
> >
> > We believe we have addressed all the issues raised by the reviewer. Could the reviewer please let us know if there are any further questions/concerns?

---

### Official Review · Reviewer_mBXz · 2021-10-28

**Correctness:** 2
**Technical Novelty And Significance:** 3
**Empirical Novelty And Significance:** 3
**Recommendation:** 5
**Confidence:** 4

**Main Review:**

The paper considers the following factorisation of the the generative model $p(\mathbf x, \mathbf c, \mathbf s) =  p(x|s, c)p(s|c)p(c)$ in which the continuous components are a function
of the categorical, where the categorical variable captures different (discrete) classes within the data and the continuous component explains variations within them.

The paper is mostly easy to read, though there are some points that require clarifications. It also comes with a quite extensive, though not complete, set of experiments and analysis.

My main issue is that the baselines against which the paper compares do not exhibit the generative structure that is given above; both of them, JointVAE and CascadeVAE, assume that
the categorical and discrete components are independent. This makes it difficult to assess whether the merits of the paper come from the factorization of the generative model that
assumes the dependence of the continuous from the categorical component or whether it comes from the concensus mechanism. Such factorizations have been quite extensively discussed
elsewhere, for example in Lavda et al, Data-dependent conditional priors for unsupervised learning of multimodal data, Entropy, 2020, where the authors discuss how such a factorisation
actually protects from mode collapse, as well as a number of other issues relevant to the discussion in the current paper, such as how the different categories are used through entropy
regularisers that naturally appear within the initial objective. Since the base level factorisation over which the present paper builds the consencus model is the same, it would have
been very easy to include it as a considerably more relevant baseline in order to demonstrate whether the benefits come from the concensus approach as well as include a discussion of
differences from Lavda et al.

One more point that I have missed concerns the claim of the paper that the multitude of arms allows to deal with cases where the class distribution is not uniform, is this something that is demonstrated in section 3.2? If yes, I did not understand what is it that makes the model robust to non-uniform $p(\mathbf c)$. Related to that are the suggestions to use 2 arms when the distribution of $p(\mathbf c)$ is uniform.


Details:

In eq~3 which gives the variational loss for arm $\alpha$ the term $\mathbb E_{q(s_a|c_a,x_a}[D_{KL}(q(c_a|x_a)||p(c_a)]$ should probably be simplified to just the KL
term $D_{KL}(q(c_a|x_a)||p(c_a))$, since the distributions in the latter do not depend on the $s_a$ variable with respect to which the expectation is taken. See for example
the respective derivation in Lavda et al, 2020.

In figure b) the graphical model describes the generative model $p(x|s, c)p(s|c)p(c)$ however what is given just bellow seems to be the inference model even though $p$ is used instead of $q$.

I have a bit of a problem conceptualising $q(c|x_1,...,x_A)$, how is this posterior instantiated concretetly? or is it only defined implicitly through equation 2? After going through the proof
of proposition 1. in the appendix this is clear.

Proposition 1. says that as the number of arms/experts increases the likelihod of the true category will also increase.

Proposition 1. :

* what is the difference of $x_i \sim p(x|m)$ and the $q(x|m)$ that appears in the expectation? $p(x|m)$ never appears in the derivation in the appendix. Is $q(x|m)$ meant to denote that
we randomly draw an $x$ and $p(x|m)$ the noisy versions of it, in which case it would probably be more appropriate to have $p(x_i|x)$.  By the way this two level sampling is missing from
the derivations, though I am not sure it is needed, but I find confusing the fact that there are two types of rvs here the original $x$ and its noisy versions $x_i$, even if I get the point.
For example in eq. 5 we marginalise over $x$ a quantity that does not contain $x$ but a random variable of $x$ the $x_i$.

* appendix, eq 5: the denominator has dissapeared and repappears again in 6.


Since the method works by operating on perturbed versions of the training instance one needs to define a perturbation model. The paper proposes a generative model based on GAN-VAE where
the main desiredata is to perturbe the instances while not alterning their latent categorical code, the non-observed class.

When computing the distance over the categorical values $c_a$ it seems that the paper considers that $c_a \in \mathcal S^K$, i.e. they belong to the probability simplex. However $c_a$ is not a probability
vector but a sample from such a categorical distribution, which should normally be a one hot-encoded vector. Which is the case? i.e. are the distances computed over the samples or over the
respective probabilities? How relevant is this?


Experiments:
* the paper said that when the distribution of classes is uniform we should use a 2-arm (dSprites and MNIST), how is this motivated? If I understood correctly the theoretical results, these show that when we increase the number of arms we will have a larger log posterior for the true categorical variable, but I am not sure I saw a relation to the uniformicity of the categories.

* In the real world experiments, figure 3, how are the categories (i.e. the columns) ordered? I presume the ordering of the three different algorithms is not really comparable and what really matters
is how these align with the real cell types. I am just curious is the orderning based on the largest values that one gets in the diagonal of the confusion matrix?

* In the same figure, it seems that the discrete compoments that the proposed method uncovers are in agreement with the hierarchical clustering of the cells. So what we have here are two
unsupervised methods the results of which are in agreement. I guess that the hierachical clustering results have been validated by the domain experts in (Tasic et al., 2018; Yao et al., 2021),
I am just wondering how to approach such an evaluation where there is no real ground truth. Of course one could say it is a good thing that two rather different methods agree on how they
cluster cells.

* The evaluation in figure 4 where a given continuous latent variable varies with the discrete component makes intuitive sense. The figure shows that the profiles produced by cpl-mixVAE varies
smoothly within a given category and look rather different between the two categories. This is not the case for JointVAE, where the categorical component seems to bring no structure, the two different
categories have rather similar profiles. The later is something to be expected, as also noted by the authors, since JointVAE assumes independence of the categorical and discrete component an assumption
that makes it not the most appropriate baseline. On the same time, other than saying that intuitively the results of clp-mixVAE look good it is hard to make an additional comment, because this requires
quite some expertise on the biology side.


**Summary Of The Paper:**

The paper presents a consensus model over a base-level VAE model that has categorical, $\mathbf c$, and continuous, $\mathbf s$ latent components. The base-level model is instantiated a number of times (arms) where given a training instance $\mathbf x$ , its arm is fed with perturbations $\mathbf x_i$ of the instance. The different arms are constrained so that they
assign the same category to the different $\mathbf x_i$ perturbations. The main motivation for such a consensus constraint is, if I am not mistaken, that it avoids the mode collapse
problem. In addition, accroding to the paper, the suggested approach can handle situations in which we have very different number of training instances in the different categories.



**Summary Of The Review:**

My main issue is that as the paper is it is difficult to assess whether the strength of the method comes from the concensus mechanism or whether it comes from the factorisation. In addition in the discussion about the number of arms being a strength when the distribution of the categorical factor is not uniform I miss the justification.

---

> ### Author Response · Authors · 2021-11-13
> **Response to reviewer mBXz - Method & Theory**
>
> We thank the reviewer for their time and feedback. First, we address the reviewer’s main concern on dissecting the main reason behind the strength of the method:
>
> - The main contribution here is using collective decision making to improve mixture representation learning in a fully unsupervised fashion. We theoretically showed that the coupled multi-arm architecture enhances accuracy of the inferred factors without requiring any priors on the relative abundances of categories, and that is not something achievable only by factorization, which is a small modification of the graphical model compared to the previous studies. In Propositions 1 and 2, we showed that applying the consensus constraint improves the categorical assignment in the mixture model.
> Additionally, in section 4.3, we have indeed provided the corresponding ablation studies and empirically showed the impact of the proposed A-arm framework under different training settings, including cpl-mixVAE$(\mathbf{s} \not\mid \mathbf{c})$, which is a cpl-mixVAE model in which the state variable is independent of the discrete variable. Our results (Supplemental Table S1) show that cpl-mixVAE$(\mathbf{s} \not\mid \mathbf{c})$ with ACC: $79.63$ performs significantly better than both JointVAE with ACC:$68.99$ and CascadeVAE with ACC: $74.83$. These experiments show that the merits of the paper come mainly from the consensus mechanism for the datasets considered here.
>
> - **Robustness to non-uniform p(c):** This stems directly from the fact that, unlike previous literature, our approach does not assume a prior for $p(\mathbf{c})$. Instead, the different arms act as surrogate priors for each other through the consensus mechanism. This is a main reason why our model outperforms the baselines significantly for the non-uniform scRNA-seq datasets while the difference is smaller for the MNIST and dSprites datasets, both of which have uniformly abundant clusters.
>
> - **Eq(sa|ca,xa[DKL(q(ca|xa)||p(ca)]:** yes, it can be simplified as you suggested, and we also used the simplified expression for the cpl-mixVAE loss function, in Eq 7. To address the reviewer concern, we will simply this term.
>
> - **“...Figure (1.b), graphical model describes generative model p(x|s,c)p(s|c)p(c), p vs. q":** as the reviewer mentioned, the graphical model only represents the generative process, $p(\mathbf{x}_a|\mathbf{c}_a, \mathbf{s}_a)$; the colorful blocks only illustrate the assumed statistical dependency between $\mathbf{c}_a$, $\mathbf{s}_a$, and $\mathbf{x}_a$ variables, not the inference process. We revised the figure to avoid any further confusion.
>
> - Proposition 1 is about the inference process in the $A$-arm VAE framework. Then we used the $q$ notation for all probability distributions. Also, in this proposition, it is assumed that $q(\mathbf{x}_i|\mathbf{x}, \mathbf{m})=q(\mathbf{x}_i|\mathbf{m})$, which corresponds to ideal type-preserving data augmentation. However, in Remark 2 (Supplement Section A), we studied the under-exploration scenario in data augmentation, in which the proof follows in the same way except the augmented samples are no longer conditionally independent, using $q(\mathbf{x}_i|\mathbf{x}, \mathbf{m})$.
>
> - **Supplement Eq. 5:** Eq. 5 expresses the log-likelihood and Eq.6 denotes the expected posterior for the $m$-th category. Could the reviewer please let us know if this did not address the reviewer’s concern?
>
> -  **one-hot representation:** Categorical variables are not encoded as one-hot vectors. To sample from $q(\mathbf{c}_a|\mathbf{x}_a)$, we used the Gumbel-softmax distribution (Section 4, first paragraph), which uses $\tau$, temperature parameter that controls how closely the samples approximate discrete one-hot vectors. The temperature values for each experiment are reported in the supplement, Section L. Also, the distances are computed over the posterior probabilities, not the samples.

---

> > ### Author Response · Authors · 2021-11-27
> > **Follow up**
> >
> > We believe we have addressed all the issues raised by the reviewer. In particular, we believe we have carefully addressed issues related to "Correctness" and the paper does not have incorrect claims.
> > We would like to respectfully ask the reviewer to reconsider their scores.
> > We are also happy to address any further questions.

---

> ### Author Response · Authors · 2021-11-13
> **Response to reviewer mBXz - Experiment**
>
> - **Uniform distribution and 2-arm:** In Corollary 1, supplement section A, we showed that for a uniform prior on the discrete factor, one pair of VAE arms ($A$= 2) is sufficient to guarantee a true categorical assignment on expectation. This is also briefly discussed in the paragraph after Proposition 2. We will revise this paragraph to emphasize this aspect more clearly.
>
> - **“as the number of arms/experts increases the likelihood of the true category will also increase”:** This is indeed a key point of our contribution. This statement is true only under the type-preserving augmentation assumption. That is why we also experimentally demonstrated that our data augmenter can achieve almost perfect type-preserving augmentations in the supplement, Sections F.1 and F.2.
>
> - **"is the orderning based on the largest values that one gets in the diagonal of the confusion matrix?":** Yes, the ordering is essentially based on the largest values that one gets in the diagonal of the confusion matrix: the ordering on the y-axis is based on the proposed dendrogram for neuronal types in the mouse brain (Tasic et al., 2018, Yao et al., 2021). The ordering on the x-axis (our categorical variables) is based on the minimum weight matching algorithm to uniquely assign one category to one neuronal cell type (Section 4, first paragraph).
>
> - **“I am just wondering how to approach such an evaluation where there is no real ground truth. Of course one could say it is a good thing that two rather different methods agree on how they cluster cells”:** We agree with the reviewer’s observation and evaluation. In addition to the continuous variable an analysis and providing a confirmation through a principled variational approach, a further use of our method is inferring improved cluster memberships, particularly in high dimensional settings. To demonstrate this, we have now added an additional section (Section M) and a figure (Fig. S14) in the updated supplementary file. By quantifying the intra vs inter cluster distances (Silhouette score), Fig. S14 demonstrates that our method improved cluster memberships (identifying more separable clusters) in high dimensional settings.

---

### Official Review · Reviewer_J8W6 · 2021-10-29

**Correctness:** 2
**Technical Novelty And Significance:** 3
**Empirical Novelty And Significance:** 2
**Recommendation:** 5
**Confidence:** 5

**Main Review:**

Update: I have adjusted my score upwards.  The reviewers have address some of my concerns.  I still believe that the interplay between the  VAE-GAN and the clustering method is not explored well-enough, and that complex interplay can drastically impact the results of the approach.

I have issues with the experiments, justifications, and theory in this manuscript.

First, my major issue with this manuscript is that the proposed approach is not truly an unsupervised method, as is claimed.  It uses some supervised information in order to make its decisions.  To see this, consider the usage of “type-preserving data augmentation that generates independent and identically distributed copies of data while preserving its categorical identity.”  The generation process maintains the label; hence, you are telling the neural network that 2 samples from the same class belong in the same cluster.  Thus, because the data augmentation system is using full data distributions, despite not explicitly using the label, I would argue strongly that supervised information is leaking into your system.  The reason that this data augmentation does not help JointVAE$^\dagger$ like it helps cpl-mixVAE is that in the JointVAE the system is not told that these should exactly match, so the supervised information does not leak into the JointVAE system.

As such, if you are going to use existing clusters, I need to understand the benefits of using this approach rather than a full supervised approach before any consideration of acceptance.  This is especially true as the quantitative metrics are all supervised evaluation criteria.  Since your method is getting some supervised information, it is unsurprising that it does better.  At the same time, it does much worse than basic supervised methods, including logistic regression on MNIST.

If you are going to push this as an unsupervised method, you should explain how it could be used in an unsupervised fashion.  For example, how well does it represent the data in an unsupervised fashion.  Can you use it for model selection?

Theory issues:

Both proposition 1 and proposition 2 require that you know the true $m$ to estimate $m$.  This is not unsupervised theory, and it is misleading to call it that.  Second, the interpretation that the expected log posterior increases with the number of arms is misleading.  This method artificially inflates the posterior and isn’t adding information to the system.  If you are given a new sample where you don’t know the cluster, and does not improve the correct assignment rate in the Bayes optimal case.  This theoretical section needs an increased discussion about these issues.

Minor complaints:

It is confusing to introduce c and s as independent in (1) and then immediately couple them in (3) without discussion.  Please refine this transition.

Expectation on the last term in (3) is on the wrong distribution.

It is unclear how the distance between the distribution on c in (6) is related to the Aitchison distance used later.  Please provide the relationship.

Additional explanation of 3.3 is necessary.  It seems like the primary benefit is to push it away from trivial solutions, and the authors should expand on how they do that.

In the ablation study, I would like to understand the impact of the novel distance better.


**Summary Of The Paper:**

This manuscript proposes cpl-mixVAE, which is a novel VAE formulation that attempts to improve clustering models by adapting ideas from consensus clustering.  The paper proposes a new model structure and a novel distance to help encourage improved model representations.

**Summary Of The Review:**

I find the idea of including consensus clustering in a VAE an interesting and potentially useful idea to explore, but the theory explicitly suggests that this model only helps in the supervised case, not an unsupervised case.  Additionally, I would argue that supervised information leaks into cpl-mixVAE through the data augmentation, as is the explicit goal, and it is unfair to compare to purely unsupervised methods on only supervised metrics.

---

> ### Author Response · Authors · 2021-11-13
> **Response to the main concerns**
>
> We thank the reviewer for their time and feedback. However, we respectfully disagree with the reviewer’s main arguments, and believe that these are due to confusions. Here, we address them to the best of our understanding:
>
> - The main focus of our paper is on continuous-discrete (mixture) unsupervised latent representation of cellular diversity, not just the clustering task. The proposed method indeed represents an unsupervised approach as the only inputs are the raw measurements and the dimension of the latent factors $|\mathbf{c}|$ and $|\mathbf{s}|$. The $A$-arm VAE framework does not have access to cluster labels, their distribution or any other supervision. The framework internally uses data augmentation to make multiple copies of the given samples, without knowing their cluster identities. The proposed cpl-mixVAE model constrains VAE arms to cooperate and agree on the discrete variable, again internally, without ever receiving supervision on the cluster identity of the samples. Consensus among arms is not reached via external information of any kind; neither individual cluster labels, nor pairwise cluster affinities, nor any distribution thereof.
>
> - **“unsupervised information is leaking into your system”:** The key point is that the “leak” is an internal signal; the system creates its own self-supervision through the proposed coupling paradigm.
> The augmented samples are achieved within the framework through a VAE-GAN approach, which is trained to learn the noise characteristics within each class, again without using any cluster information. We respectfully ask the reviewer to revisit this part of the review.
>
> - **“it is unfair to compare to purely unsupervised methods”:** We again stress that all the methods have access to the exact same information. No other information is ever provided to any of the methods reported in this paper (cpl-mixVAE, JointVAE, CascadeVAE). The main reason for the superior performance of cpl-mixVAE is its use of the proposed constrained optimization (Eq. 6), which is justified in Propositions 1 and 2.
>
> - **“... you are going to use existing clusters”:** To further clarify, no existing cluster information other than the total count, |c| is ever provided to any of the methods.
>
> - **“you are telling the neural network that 2 samples from the same class belong in the same cluster”:** No such information regarding a pair of actual samples is ever provided, which would indeed have constituted extra information. Rather, different versions of a single sample are produced internally, without access to any external information.
>
> - **Can you use it for model selection?** Our approach can indeed be used for model selection, for instance, in the supplement Section J, we investigated the performance of cpl-mixVAE for different cardinalities of the categorical variable (Fig.  S9).  Our results demonstrate that while 1-arm VAE suffers from sensitivity to $|\mathbf{c}|$, cpl-mixVAE is more robust in encoding the discrete variability, and the log-ratio measure is maximized for the true number of clusters.
>
> - **“Both proposition 1 and proposition 2 require that you know the true m to estimate m”:** This is not correct. These propositions merely calculate the expected log of conditional posterior probability of correct assignment to justify, in the presence of a sufficient number of arms, the inferred categorical variable will agree with the true label in expectation, as theoretical guarantees for our inference algorithm. However, the model does not use label information in any way in inference.
> We would be happy to further clarify if this response did not address the reviewer’s concern.
>
> - **“you should explain how it could be used in an unsupervised fashion”:** Our experiments already demonstrate applications of our method to real-world problems, concretely, in single-cell omic studies. For instance, we systematically relate -- for the first time -- genes to jointly inferred continuous variability. It can also infer the hierarchical relationships among samples (Fig. S10) and improved cluster memberships (newly added Supplementary Fig. S14) for high dimensional datasets.

---

> > ### Comment · Reviewer_J8W6 · 2021-11-18
> > **I remained unconvinced that supervised information is not leaking in; please provide additional evidence for this claim**
> >
> > **On supervised information leaking into the system**:
> >
> > The theory, to my understanding, is dependent on knowing the correct cardinality of the clustering and being able to produce multiple samples from the same class.  This is noted by the authors as “when the augmentation is type-preserving, by definition, $q(x_i|x_j,c) = q(x_i|c)$.”  I understand the author’s response, “the key point is that the ‘leak’ is an internal signal.” As the author’s note, “the system creates its own self-supervision through the proposed coupling paradigm. The augmented samples are achieved within the framework through a VAE-GAN approach, which is trained to learn the noise characteristics within each class, again without using any cluster information.”  It is $unclear$ the extent to which this is true.  The VAE-GAN approach has a lot of tuning parameters that impact performance, and the authors choose, whether explicitly or implicitly, parameters that very effectively captured the label information.  As they note, their result “demonstrates that the augmenter preserves the label information (type) for 96.58% of the augmented samples” in the MNIST case.  Likewise, Figure S3 suggests that the cluster information was used to tune and evaluate the generation system in scRNA-seq data.  Maybe this was not explicitly used in training of this system, but supervised information was certainly used to evaluate the appropriateness of this data generation procedure.  Figuring out whether this type-preserving generation is appropriate without known clusters seems very difficult, and is not sufficiently discussed.
> >
> > I do like the idea of creating this self-supervised type-preserving network, and I am open to the authors effectively convincing me that the proposed approach is in fact not leaking supervised information into this system.  In order to do this, I suggest additional experimentation on (a) the sensitivity to the tuning of this step and (b) an increased discussion on how this would be evaluated when you don’t know the labels (the actual unsupervised case).
> >
> > I did understand that only the number of clusters, |c|, was used in the experimental section.  That is still a lot of information provided to a system, and is nontrivial to know. I appreciate the authors pointing me towards the study in Appendix J, which I had missed. Figure S9 is a nice figure showing that this system can capture the number of clusters effectively in MNIST.  I would suggest discussing that more in the main draft.  However, there is a complex interplay between the VAE-GAN step and this evaluation.  Following up on the prior point, I would need to understand how reliant these results are on a well-tuned VAE-GAN.  For example, the chosen parameters of the VAE-GAN effectively capture that there are 10 clusters in MNIST, which the authors at a minimum evaluated.  How sensitive is this approach to a suboptimal VAE-GAN?

---

> > > ### Author Response · Authors · 2021-11-21
> > > **Response to the remaining concerns (II)**
> > >
> > > - **“I suggest additional experimentation on (a) the sensitivity to the tuning of this step” … “How sensitive is this approach to a suboptimal VAE-GAN”:** We have reported multiple ablation studies in the supplementary file that we believe would address the reviewer's concerns.
> > >
> > > ·       In Section G, we analyzed the sensitivity of the proposed coupled framework to the coupling hyperparameter and showed that cpl-mixVAE's performance is adequate for different values of the coupling factor (Figure S6).
> > >
> > > ·       In Section J, Table S1, we reported cpl-mixVAE$^*$, cpl-mixVAE in which VAE arms are not independent and all networks' parameters are shared. Our results show that, in the presence of augmentation, sharing all network parameters impairs the mixture representation.
> > >
> > > ·       In Section J, Table S1, we reported cpl-mixVAE$^a$, a cpl-mixVAE model that does not use the proposed VAE-GAN data augmentation, and instead uses a set of simple affine transformations e.g., rotation. This result shows that, even in the absence of optimal (ideal) type-preserving augmenter, the model still outperforms the comparable single-VAE approach.
> > >
> > > ·       In Section K, we studied the Robustness of type-dependent latent factors in cpl-mixVAE and our results show consistency for the inferred continuous variable, across multiple randomly initialized runs.
> > >
> > > - **"an increased discussion on how this would be evaluated when you don’t know the labels”:** We attempted to address this important aspect by focusing on the single-cell application, where there is no consensus (gold standard) annotation of cell types in the field, and it was one of our motivations in pursuing this problem. We also included the Silhouette analysis which indeed presents how well our inferred labels describe the underlying high-dimensional dataset, compared to the existing taxonomy. To address the reviewer’s concern, we edited the manuscript and the supplementary file to emphasize that (i) the generative model learns to generate samples in the vicinity of a given sample in the latent space, without access to any label information in any way, (ii) we call such generation of samples type-preserving not because label information was utilized during generation or label preservation is guaranteed, but because ‘similar’ samples often belong to the same cluster and this nomenclature may help relate the implemented model and the theory, (iii) the generative model does not have access to cluster count, either.
> > >
> > > - **“I did understand that only the number of clusters, |c|, was used in the experimental section. That is still a lot of information provided to a system, and is nontrivial to know.”:** Virtually all parametric unsupervised clustering methods use the cluster count or another surrogate variable, e.g., neighborhood distance, as a hyperparameter. For instance, classical unsupervised clustering algorithms, such as K-means, DBSCAN, GMM, all use such a parameter. Moreover, in the proposed cpl-mixVAE framework, although we need to pre-define the cluster count as a hyperparameter of the deep network, we demonstrated that the model does not need to know the true cluster count and can recover meaningful clusters even with incorrect cluster counts. (Figures S9 and S10). As mentioned by the reviewer, Figure S9 demonstrates that the true number of clusters can be recovered via model selection. To address the reviewer‘s concern, we will point to the study in Figure S9 more prominently in the main text.

---

> > > > ### Comment · Reviewer_J8W6 · 2021-11-22
> > > > **Thank you for the revision and details; I still recommend additional exploration**
> > > >
> > > > Thank you for providing the additional feedback.  I did adjust my score.
> > > >
> > > > I had missed Table S1, which to my knowledge was not referenced or discussed in the main draft.  I would encourage the authors to think about how to succinctly present the conclusions to the main draft.
> > > >
> > > > The results in Table S1 are promising, but I am unsure how general they are.  The simple rotation, while not optimal, seems a perfect type-preserving augmentation.  I would really like to understand the impact of an imperfect type-preserving generator on the system. What would the impact be if it was highly suboptimal?  For most data problems, I would expect this to be highly suboptimal.  I would recommend this for future exploration.

---

> > > > > ### Author Response · Authors · 2021-11-23
> > > > > **Response to the remaining concenrs**
> > > > >
> > > > > We are glad to see our response addressed the reviewer’s concerns about the unsupervised learning. Here, we attempt to address the remaining concerns.
> > > > >
> > > > > - Section 4.3 (Ablation Studies), in the main text, prominently refers to Table S1, and describes a summary of most of the supporting experiments we conducted. Given the additional page in the camera-ready submission, we will include this table in the main text.
> > > > >
> > > > > - **perfect vs sub-optimal augmentations:** In the proposed multi-arm VAE framework, a perfect data augmenter is a generative model that fully learns within-cluster variations. Conventional transformations such as rotation are not perfect augmentations, because they do not capture the entire underlying variability for each digit. Additionally, under the rotation transformation, there are cases where the type identity also changes. For instance, certain '4's can be confused with '9's when they are rotated in the clockwise direction. Therefore, it wouldn't be accurate to consider small-angle rotations as perfect augmentations, although they are not far from it.
> > > > >
> > > > >    While in theory (propositions 1 and 2), A-arm VAE framework relies on a perfect type-preserving data augmentation, in Supplementary Section A, we provided a theoretical explanation for sub-optimal data-augmentation, i.e. Remark 2. This remark is referred to on page 4, under the “Arms observe non-identical copies of samples” sub-section. In general, the perfect type-preserving augmentation is an idealization that is very hard to achieve as soon as the underlying dataset is not a toy dataset. That is why we devised a VAE-GAN module to capture as much of the richness and complexity of the data as possible. As we already reported, this module achieves 96.58% of "type-preservation" in the case of the MNIST dataset. Therefore, this module is not perfect, either.
> > > > >
> > > > >    Taken together, the theoretical and experimental results suggest that the method should not perform well at all if the type-preserving generator was "highly suboptimal". Intuitively, if augmented data is essentially noise, what would be there to learn? Similarly, if augmented data swaps digit identities, what would be there to learn? Here, the proposed VAE-GAN module represents a general-purpose augmenter, whose success has been demonstrated on 4 different datasets (image and non-image datasets). Not only the size and scale of these datasets are different, their characteristics also vary considerably. More specifically, our experiments and ablation studies show that this augmenter performs really well for complex scRNA-seq datasets, which is our main application in this manuscript.

---

> > > > > > ### Author Response · Authors · 2021-11-27
> > > > > > **Follow up**
> > > > > >
> > > > > > We realized that the "Correctness" score has not been updated. We believe the paper does not have incorrect claims and we have addressed the reviewer's concerns in this regard. We would like to respectfully ask the reviewer to reevaluate their scores.

---

> > > ### Author Response · Authors · 2021-11-21
> > > **Response to the remaining concerns (I)**
> > >
> > > We thank the reviewer for the feedback and appreciate the time spent considering our response. In the following, we aim to address the remaining concerns.
> > >
> > > - **VAE-GAN data augmenter:** The proposed VAE-GAN model learns to generate samples in the vicinity of a given sample (in the latent space). The approximate type-preserving phenomenon is merely a consequence of this property. The VAE-GAN model does not have access to either the labels or the cluster count.
> > > In supplementary section F, we have explained the VAE-GAN objective function and discussed that how each optimization term (supplement, Eq. 70) contributes to the data augmentation process without using the cluster information. The proposed loss function is independent of $|\mathbf{c}|$ and $|\mathbf{s}|$ and uses a triplet loss that prevents the network from generating identical samples and a probabilistic distance measure that encourages the network to place the original and noisy samples close to one another in the latent space (please see section F).
> > >
> > > - **“As they note, their result ‘demonstrates that the augmenter preserves the label information (type) for 96.58% of the augmented samples’ in the MNIST case. Likewise, Figure S3 suggests that the cluster information was used to tune and evaluate the generation system in scRNA-seq data”:** As explained above, the data augmenter does not have access to cluster information at all (neither labels nor cluster count). Supervised information was never used to train (tune) the generative model.  Figures S2 and S3 and the reported number for MNIST are post-hoc qualitative and quantitative evaluations for the reader to gain insight about the proposed VAE-GAN model as a type-preserving data augmenter and to see how well the generator keeps the categorical component unchanged, without knowing about the cluster label of the given sample at the time of training.
> > > Additionally, Figure S3 show a simple 2-dimensional visualization of gene expression based on a simple autoencoder, not cpl-mixVAE. It is only reported for the reader to compare the original single cell measurements with the augmented samples.
> > >
> > > - **“The theory, to my understanding, is dependent on knowing the correct cardinality of the clustering and being able to produce multiple samples from the same class”:** It is not correct that the theory depends on knowing the true number of clusters. Neither the derivation of the coupled variational objective nor the propositions know how many clusters should be inferred. Propositions 1 and 2 state that given sufficient number of arms in the multi-arm VAE framework, the maximum of expected log-conditional probability of the categorical variable, is achieved at the category where the augmented samples come from. It should be noted that the theory justifies the performance of the model at time of inference, not training. During training, the model never has access to the true categorical information.
> > >
> > > We also note that while we need to pre-define the cluster count as a hyperparameter of the deep network in cpl-mixVAE, we demonstrated that the model does not need to know the true cluster count and can recover meaningful clusters even with incorrect cluster counts (Figures S9 and S10).
> > >
> > > It is true that the theory depends on the (unsupervised) ability to produce multiple samples from the same class, as this assumption is clearly stated in the propositions, and the development in the main text and the supplement. (Please see the definition of multi-arm framework in Sec. 3.1.)
> > >
> > > -**“supervised information was certainly used to evaluate the appropriateness of this data generation procedure”:** To avoid any potential confusion, we also wish to make a more general statement on evaluation: our main aim in this study is probabilistic mixture modeling in which we jointly learn interpretable discrete variables, $\mathbf{c}$, and continuous variables, $\mathbf{s}$. That said, how should one evaluate success here? How to know whether the method performs well – after training -- in the context of mixture representation learning i.e., $q(\mathbf{c}|\mathbf{x})$ and $q(\mathbf{s}|\mathbf{c}, \mathbf{x})$? To evaluate representations, we used six metrics, (1) accuracy (Table1 and Figure 3), (2) Disentanglement score (Table 1), (3) $D_{KL}(q(\mathbf{c}) || p(\mathbf{c}))$ (Table 1), (4) latent traversal (Figures 2 and 4), (5) reconstruction error (Table S1), and (6) Silhouette scores (recently added, Figure S14). These represent common measures in representation learning. Here, only in measures (1) and (3) we use cluster information, $p(\mathbf{c})$, and that does not mean that the unsupervised learning paradigm is violated, since these are strictly post-hoc analyses that give us insight about the representation and demonstrate the pros and cons of the method compared to existing methods or the ground truth, when available.

---

> ### Author Response · Authors · 2021-11-13
> **Response to the minor concerns**
>
> - **“It is confusing to introduce c and s as independent in (1) and then immediately couple them in (3) without discussion”:** The ELBO in Eq. 1 is introduced to discuss the conventional single mixture VAE used in JointVAE and CascadeVAE and its issues (Section 2). In Section 3, we introduced our variational loss (Eq. 3) that acknowledges the dependency between $q(\mathbf{s}|\mathbf{x})$ and $q(\mathbf{c}|\mathbf{x})$, which is missing in Eq. 1 (discussed at the end of Section 2).
>
> - **“Expectation on the last term in (3) is on the wrong distribution”:** Nothing is wrong about that last term and expectation. We carefully checked our derivation, which is also provided in the supplement Section B. Since the distributions in the $D_{KL}$ term are not functions of $q(s)$, we can simplify and rewrite it as $ D_{KL}\left(q(\mathbf{c}|\mathbf{x}) \| p(\mathbf{c}) \right) $. But, nothing is wrong about that last term and expectation.
> If this response or the derivation in the supplement does not address the reviewer’s concern, could the reviewer please explain why they think the expectation is on the wrong distribution?
>
> - **“It is unclear how the distance between the distribution on c in (6) is related to the Aitchison distance used later”**:  In the paragraph right after Eq. 6, we explained that to train the $A$-arm VAE model, we relaxed the optimization and instead of maximizing the loss in Eq. 6 , we maximize the loss in Eq. 7, which is a function of the distance between a pair of categorical variables. To address the reviewer’s concern, we will modify Section 3.2 to better clarify this connection.
>
> - **Section 3.3:** As mentioned by the reviewer, in this section we proposed a novel calculation of perturbed distances in the simplex as a surrogate dissimilarity measure, to avoid the mode collapse problem. Due to the page limitations, we explained all details regarding Aitchison geometry and the derivations of the distance in the supplement Section E. If the reviewer has further specific concerns, we will be happy to address them.
>
> - **“the impact of the novel distance”:** The novel distance in Section 3.3 is proposed to avoid trivial solutions in the simplex. In the absence of this formulation, we observed that only a few categories are actively used (mode collapse). We will add these results in the supplement file.

---

### Official Review · Reviewer_uWQ6 · 2021-11-01

**Correctness:** 3
**Technical Novelty And Significance:** 2
**Empirical Novelty And Significance:** 2
**Recommendation:** 5
**Confidence:** 4

**Main Review:**

Post-rebuttal:
The authors' feedback clarify some of my concerns. but I'm still not so convinced by the experiments. The authors argue the model is not suitable for SVHN/cifar10 due to their complexity and this makes it unclear whether the model is applicable/generalizable for the real-world datasets that are either large scale or present extremely challenging backgrounds. Achieve SOTA ACC is definitely not necessary, but I would assume the proposed model has competitive results, however, the preliminary results shown in appendix seem quite weak. Therefore, I still tend to keep my original rating.

For the paper strength:

-- The latent variable with mixed representation is an important model and worth exploring.

-- Not familiar with the bio-literature, but the experiments and applications on RNA-data look interesting.

For weakness:

-- The motivations and descriptions of the model need to be improved. Some places are unclear and confusing. For example. the paper emphasized that the work is fully unsupervised and proposes the type-preserving data augmentation. However, since each arm should receive samples that share the same underlying categorical factor, does this requirement impose some weak-supervision signal?  i.e., we have to ensure that each arm observes the samples from same category, but for fully unsupervised setting, we don't know that. Could authors further elaborate on this point?

-- The experiments seems not sufficient. (a) Since more arms potentially means more encoder/decoder pairs (i.e., more parameters involved), its better to also include the model complexity comparison with the baselines. (b) The scalability of the model is not clear since the experiments are only done on synthetic/gray-scale datasets, what about on svhn or cifar10? For these slightly complicated benchmarks, how does the model perform? How to select the number of arms? Does the model capable dealing with large-scale datasets?

**Summary Of The Paper:**

The paper proposes to learn latent variable model with mixed discrete and continuous latent variables. Specifically, the framework utilizes multiple pairwise-coupled autoencoding arms to learn shared categorical variable. The experiments on dSprites, MNIST and RNA-seq datasets verify the effectiveness of the model.

**Summary Of The Review:**

The major concern goes to the experiment setting. The benchmarks are too simple to show the potential of the model and the baselines compared can be more up-to-date. I lean towards reject and may reconsider my rating based on authors' feedbacks.

---

> ### Author Response · Authors · 2021-11-13
> **Response to reviewer uWQ6**
>
> We thank the reviewer for the comments. In the following, we aim to address all of the concerns:
>
> - **Weak supervision:** The proposed framework does not use any prior information about the cluster identity. The weak-supervision, as interpreted by the reviewer, is an “internal” signal that is provided by the data-augmenter, which does not use any “external” information. While training, the proposed framework receives a data point, x, without knowing its cluster identity, generates $A$ non-identical copies of it, and learns an $A$ tuple of $(\mathbf{s}_a, \mathbf{c}_a)$, where  $\mathbf{c}_a=\mathbf{c}_b=\dots$ . Everything is inferred from within the model.  Could the reviewer please let us know if this explanation did not sufficiently address the reviewer’s concern?
>
> - **“we have to ensure that each arm observes the samples from same category”:**  We do not ensure this during training. The proposed framework internally generates multiple noisy copies of the given samples, without knowing their cluster identities. Consensus among arms is not reached via external information of any kind. No information regarding a pair of actual samples is ever provided, which would indeed have constituted extra information. Rather, different versions of a single sample are produced internally, without access to any external information.
>
> - **Experiment – model complexity:** In Table 1 (page 6 of the main manuscript), we reported the model complexity (additional computational cost due to having more parameters) as computational efficiency, i.e., number of iterations per second to compare the training complexity of the cpl-mixVAE against baselines.
>
> - **Experiment – scalability:** The $A$-arm framework is originally proposed for omic datasets with high dimensional discrete settings and over hundreds of thousands of cells, two of which are used in our experiments. The main reason for using the MNIST and dSprites benchmarks is to facilitate comparisons of our method with earlier ones. Additionally, the results in Table 1 actually suggest that the performance gap between cpl-mixVAE and the baselines tends to increase as the complexity of the dataset increases while having fairly similar computational cost.
>
> - **Experiment – image datasets:** In the field of joint inference/disentanglement, we note that MNIST and dSprites are the somewhat standard benchmarks, as can be appreciated by the papers cited in the literature review and the baseline papers. Suggested datasets, i.e., svhn and cifar10 would not add much beyond the MNIST and dSprites benchmarks because both of these datasets have only 10 clusters (versus 100+ clusters in the single-cell datasets) and have almost equally abundant clusters (versus the highly divergent cluster sizes in the single-cell datasets).
>
> - **Selecting number of arms:** The number of arms has been discussed in the supplement Section A, proof of Proposition 2 and Corollary 1 and shortly in main text, Section 3 after Proposition 2.
>
> - **Experiment – baseline methods:** In the Related Work section, we cited several recent works in representation learning. Among recent studies, the baselines studied in this paper (Dupont 2018, Jeong 2019) are strong and representative of the state-of-the-art in the field of “joint discrete-continuous unsupervised representation learning”.
> We would appreciate it if the reviewer could let us know about any other strong and comparable methods.

---

> ### Author Response · Authors · 2021-11-30
> **Response to the post-rebuttal comment**
>
> We just noticed the reviewer’s post-rebuttal comment was posted as a modification in the main review section of the initial comment. Here we would like to address the reviewer’s concerns.
>
> -**“unclear whether the model is applicable/generalizable for the real-world datasets”:** Single-cell datasets are real-world datasets and an important application domain. A key observation is that the gene expression vector does not exhibit spatial structure (e.g., no shift invariance, unlike images), so many methods focusing on image data are simply not applicable here, including ACOL (Kilinc et al, 2018) in Table S2 of the supplementary section N (please see Fig. 8 of Kilinc et al.).
>
> -**“Achieve SOTA ACC is definitely not necessary, but I would assume the proposed model has competitive results, however, the preliminary results shown in appendix seem quite weak”**:  We would like to emphasize that the comparable methods here are mixture model-based approaches, e.g. JointVAE, not clustering methods. We clearly discussed in the main text and supplement that here, the goal is not image clustering. We aim to learn representations that jointly model continuous and discrete variational factors — because these factors are typically not independent of each other — using the assumed mixture probabilistic model (graphical model in Fig. 1b). Therefore, we believe concluding that the model is not general enough based on SVHN results would not be accurate.

---

### Official Review · Reviewer_giw4 · 2021-11-02

**Correctness:** 4
**Technical Novelty And Significance:** 3
**Empirical Novelty And Significance:** 3
**Recommendation:** 8
**Confidence:** 4

**Main Review:**

Strengths: The idea is quiet novel and smart. The technical formulation is appropriate, and the constraint enforcing mixture component assignment consensus is sound. The experimental results are convincing, as they include a number of benchmarks, including some challenging ones, comparison to some SOTA methods, and an ablation study that offers good insights into how and why the method works.

Weaknesses: I did not find any commendable weakness in the paper.

**Summary Of The Paper:**

The paper considers the problem of generative modeling for mixed discrete-continuous data. To this end, it introduces a novel variant of coupled variational autoencoders. Specifically, they reformulate the coupled variational autoencoder paradigm in such a way that, instead of each "arm" of the autoencoder modeling a different modality, the arms model different chunks of the same modality (which comprises a high-dimensional discrete part, in addition to a continuous part).

The key novelty of the paper is the idea of postulating a finite mixture model as the likelihood (decoder) pertaining to each arm of the autoencoder, and stipulating that all arms essentially infer similar categorical posteriors of mixture component assignment for the different chunks pertaining to the same observation. This consensus constraint is enforced on the grounds of the Aitchison geometry in the probability simplex, which avoids the mode collapse problem.


**Summary Of The Review:**

The paper is novel enough, methodologically sound, and empirically well-validated.

---

> ### Author Response · Authors · 2021-11-13
> **Response to reviewer giw4**
>
> We thank the reviewer for their time and encouraging comments.

---

### Author Response · Authors · 2021-11-23
**General response**

We thank the reviewers who have already responded to our comments. We believe we have addressed all the issues raised by the reviewers and are happy to address the remaining concerns, if any. Going through the reviews, it occurred to us that some of the studies in our supplemental file may have been overlooked. Therefore, we would like to take this opportunity to summarize our main contributions, the supplemental/ablation experiments, and the newly added analyses.

Main contributions (in the main text):
- A novel unsupervised representation learning framework (cpl-mixVAE) that jointly learns interpretable continuous and discrete variational factors (Section 3)
- Theoretical justification for superiority of cpl-mixVAE compared to single mixture VAE models (Section 3.1)
- A novel distance for the discrete representation to prevent mode collapse in the probability simplex (Section 3)
- Experimental demonstration of cpl-mixVAE outperforming comparable mixture VAE models, on both two different benchmarks datasets and -- more importantly -- two single-cell RNA sequencing datasets, each profiling over 100 neuron types (Section 4, Tabel 1, Figures 2 and 3)
- Demonstration of joint discovery of neuronal types as discrete categories and type-specific genes that regulate the continuous within-cell type variability, such as metabolic state or disease state. To the best of our knowledge, here we present the first method to address this issue with single-cell datasets in an unsupervised manner (Section 4, Figure 4)

Supporting studies (in the supplement):

- A new generative model for unsupervised type-preserving data augmentation (Section F)
- A set of qualitative and quantitative assessments of the proposed data augmentation (Section F, Figures S2, S3, S4, and S5)
- An ablation study to show data augmentation by itself does not improve the performance of the 1-arm VAE model (Section J, JointVAE$\dagger$ )
- An ablation study to investigate the impact of network architecture (Section J, JointVAE$\ddagger$)
- An ablation study to show that the model’s performance drops when all VAEs’ parameters are shared (Section J, cpl-mixVAE$^*$)
- An ablation study to further show the importance of statistical dependency between the continuous and the discrete variable on the model’s performance (Section J, cpl-mixVAE$(\mathbf{s} \not\mid \mathbf{c})$)
- An ablation study to show the proposed model performs reasonably, even under a simple affine augmentation (Section J, cpl-mixVAE$^a$)
- Sensitivity analysis for the hyperparameters (Section G)
- Robustness analysis of the continuous factors in single-cell datasets (Section K.1)
- Additional continuous traversal analyses to further show the results for more genes regulating cell activity (Section K.2)

(Newly added analyses upon consideration of the reviewers’ comments)
- Silhouette score analysis demonstrating how our approach can improve categorical label assignments when no ground truth is available (Section M)
- A new set of experiments on the SVHN dataset (Section N)

---

### Decision · Program_Chairs · 2022-01-20

**Decision:**

Reject

**Comment:**

This paper proposes cpl-mixVAE, a method for fitting discrete-continuous latent variable models based on mixture representations and a novel consensus clustering constraint. After extensive discussion, no one was willing to argue in favor of acceptance, and a majority of the reviewers felt another round of revision is needed. Ultimately, I concur that while the ideas are novel and potentially interesting, more effort is needed to convincingly demonstrate the efficacy of the method. Valid concerns were also raised regarding the claimed "unsupervised" nature of the proposed method, a claim which at the very least requires some additional context. At this point, these outstanding issues require an additional round of revision.